# Using Powder Diffraction Patterns to Calibrate the Module Geometry of a Pixel Detector

**Jonathan P. Wright ***[iD]**, Carlotta Giacobbe and Eleanor Lawrence Bright**

ESRF-The European Synchrotron, 71 Avenue des Martyrs, 38000 Grenoble, France; giacobbe@esrf.fr (C.G.);
e.lawrencebright@esrf.fr (E.L.B.)
* Correspondence: wright@esrf.fr; Tel.: +33-4-76-88-26-83

**Abstract:** The precision and accuracy of diffraction measurements with 2D area detectors depends on how well the experimental geometry is known. A method is described to measure the module geometry in order to obtain accurate strain data using a new Eiger2 4M CdTe detector. Smooth Debye–Scherrer powder diffraction rings with excellent signal to noise were collected by using a fine-grained sample of $CeO_2$. From these powder patterns, the different components of the module alignment errors could be observed when the overall detector position was moved. A least squares fitting method was used to refine the detector module and scattering geometry for a series of powder patterns with different beam centers. A precision that is around 1/350 pixel for the module positions was obtained from the fit. This calibration was checked by free refinement of the unit cell of a silicon crystal that gave a maximum residual strain value of $2.1 \times 10^{-5}$ as the deviation from cubic symmetry.

**Keywords:** distortion; correction; pixel detector; X-ray diffraction; strain

## 1. Introduction

Photon-counting 2D area detectors that are assembled from individual modules are now becoming widely available. In comparison to previous detectors based on CCD or CMOS technologies, these photon-counting detectors offer distinct advantages. There is no readout noise and very narrow point spread function, so they are very sensitive to measure weak diffraction signals as well as offering fast readout speeds for continuous acquisition. A new Eiger2 4M CdTe detector [1] was recently commissioned at the Materials Science beamline [2], ID11, at the ESRF as a part of the EBS upgrade project [3]. A novel procedure has been developed in order to create a very precise spatial distortion correction for this new detector. The instrument is used for single-crystal, powder diffraction and PDF measurements, and these are frequently extended to XRD-CT mapping methods with beam sizes down to 100 nm [4–6] to obtain 3D images of microstructures during in situ experiments. This very high spatial resolution combined with a non-destructive measurement offers unique insights into structural variation not only within crystalline but also amorphous materials [7].

One recent focus of research efforts is the reconstruction of variations of the local strain tensor within individual grains inside a polycrystalline sample [8,9]. To obtain good strain data, the detector characterization should be as good as possible. Previous efforts to measure strains accurately using a 2D area detector showed that good calibration of the experimental geometry and distortion are essential [10]. The conventional methods [11–15] for calibration of earlier CCD detectors generally fitted smooth functions (splines or polynomials) to images of grid-like targets. The production of an accurate calibration artifact is problematic because machining tolerances are difficult to reduce below about 10 μm for a >150-mm-sized object. Typical standard objects are produced by drilling holes into a metal plate or fixing metal spheres onto a plate [16], and an image is recorded showing the shadow of this artifact. Because the detection surface not in contact with the standard

object, the quality of the calibration depends on the illumination conditions as well as the machining errors. Furthermore, a calibration grid only fixes the geometry at the grid points, and this limits the corrections to only finding the low-frequency terms in the distortion. These grid-based methods are relatively insensitive to any local defects.

Individual detector modules are essentially perfect because they are produced by lithography techniques, but the mechanical assembly of several modules frequently introduces some small misalignments. The CdTe sensor material may contain some local defects (e.g., dislocations), and there are some edge effects at module and chip boundaries. A continuous calibration image is needed to detect any short-range features like these. Knowledge of the precise pixel geometry is a requirement to avoid increasing the systematic errors, and we detected some initial issues when processing powder data. Despite the use of a grid calibration, there were still observable steps in the powder ring *2θ* positions at the module boundaries.

For very accurate measurements of diffracted intensities, e.g., for charge density refinements, there are a number of challenges with pixel detectors to be overcome [17]. One intrinsic problem is charge sharing [18] between neighboring pixels so that the sensitivity of an individual pixel has some variation over the surface of the pixel itself. The centers of pixels may sometimes be more sensitive than the corners, and this effect depends on the discrimination threshold. Any future attempt to introduce corrections for this problem into an integration and scaling software will require sub-pixel calibration of the detector geometry, which is a further motivation for this work.

There are established methods for fitting the module geometry in pixel detectors that are based on single-crystal diffraction spot positions [19]. These were developed for the XFEL detectors where the modules can be moved individually, and diffraction patterns are very spotty. The module geometry is usually fixed in commercial detectors, and the experimental setup with a continuous source makes it easier to record a smooth powder diffraction image. Methods based on single crystals require a large number of spots to be indexed in order to refine the overall geometry. Our experience measuring the centroid positions of diffraction peaks has shown some limitations that come from insufficient sampling. Because the point-spread function is extremely narrow, it is sometimes problematic to have enough points in the peak to get a precise centroid. This is overcome using a powder pattern with a beam size that is larger than the pixel size. This calibration method using continuous powder diffraction rings offers access to high-frequency terms in the spatial distortion, and we have found that relatively few images are required for the module geometry calibration.

## 2. Materials and Methods

Experimental data were collected at the materials science beamline, ID11, at the ESRF using a Dectris Eiger2 4M CdTe detector (Dectris, Baden, Switzerland). This detector delivers a single image of 2068 × 2162 pixels to the user that includes masked values for gaps or other defects. The pixel size is 75 μm square, and the CdTe sensor thickness is 0.75 mm. The detector is assembled from 2 × 4 modules that each cover 77.1 mm × 38.4 mm with a vertical gap of 2 pixels in the center of each module. Between these 2 × 4 modules, there is a 12-pixel vertical gap in the center of the detector and three horizontal gaps of 38 pixels. These gaps are shown in white on Figure 1. The aim of the experimental measurement is to collect a series of images that will give a grid-like image once they are summed together. In the current installation at the beamline, the accessible range of positions was limited by the mechanical translations. A number of authors have discussed wavelength calibration methods by translating the detector along the beam path [20–22], but here, the detector is translated in the plane of the detector surface to give different beam centers.

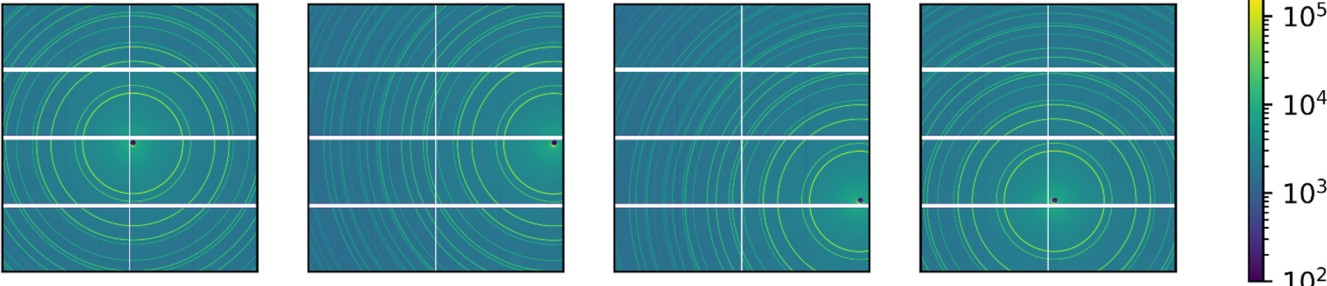

**Figure 1.** The four experimental powder diffraction patterns that were used to compute the detector module geometry. The intensity is represented on a logarithmic color scale.

A sample of ceria ($CeO_2$) with mean particle size of ~0.25 μm was mounted in a 0.5-mm diameter kapton capillary on the diffractometer axis. Two frames were measured with 10 s exposure time during a rotation of the sample over 360 degrees, and these were subsequently summed together. The signal in individual pixels in the strong low-angle peaks was ~175,000 photon counts. The X-ray energy was nominally 55 keV, and the beamsize of 0.2 mm × 0.2 mm was selected using slits in order to cover several detector pixels (that are 75 μm square) and reduce aliasing artifacts. The sample to detector distance was set to 42 cm in order to reduce effects due to the incident angle (the distance was limited by the back wall of the hutch). Figure 1 shows a rendering of the four experimental powder diffraction patterns used for these calibrations.

## 3. Results

The experimental data were processed to extract diffraction ring positions for each row and column of pixels, and these data were subsequently fitted to obtain the module geometry.

### 3.1. Imaging Processing to Extract Debye–Scherrer Ring Geometry

A background signal for each frame was estimated using a 64-pixel uniform low-pass filter with peaks being masked via a sigma clipping procedure. This estimated background signal was subtracted from the observed data before extracting peak positions. Pixels belonging to the individual modules were selected from the full image as a series of regions of interest. Each row and column of each module was treated as a separate 1D spectrum, and peaks were identified as groups of pixels that were above the estimated image background by 5-sigma (based on counting statistics) and containing more than 7 and fewer than 35 pixels in a group and containing no masked pixels. A simple center of gravity (first moment) was computed for each of these peaks. The 2D spot position was given by this computed centroid in one direction, while it corresponds to an integer value (row or column index) in the other direction. When a ring is tangential to a row or column of pixels, the width of the peak became very large, and these peaks were discarded by the upper threshold of 35 pixels. The computer code for this image processing was implemented in the python language using a jupyter notebook [23] and the numba JIT compiler [24] to speed up the peak position extraction. The total runtime for processing the four frames of 2162 × 2068 pixels was about 10 s.

Figure 2 shows the counting statistics and sampling for a single column of pixels and the low angle (111) reflection on the Figure 2a. Figure 2b highlights a small region of interest where the powder ring crosses a boundary between two detector chips. Some pixel values in this region were interpolated in the manufacturer's software via a "virtual pixel correction" method, and the columns 770–771 and 772–773 are essentially duplicates of each other. These kinds of effects are clearly seen with a continuous signal in a powder diffraction ring but are much more difficult to infer from a grid image or spotty diffraction pattern.

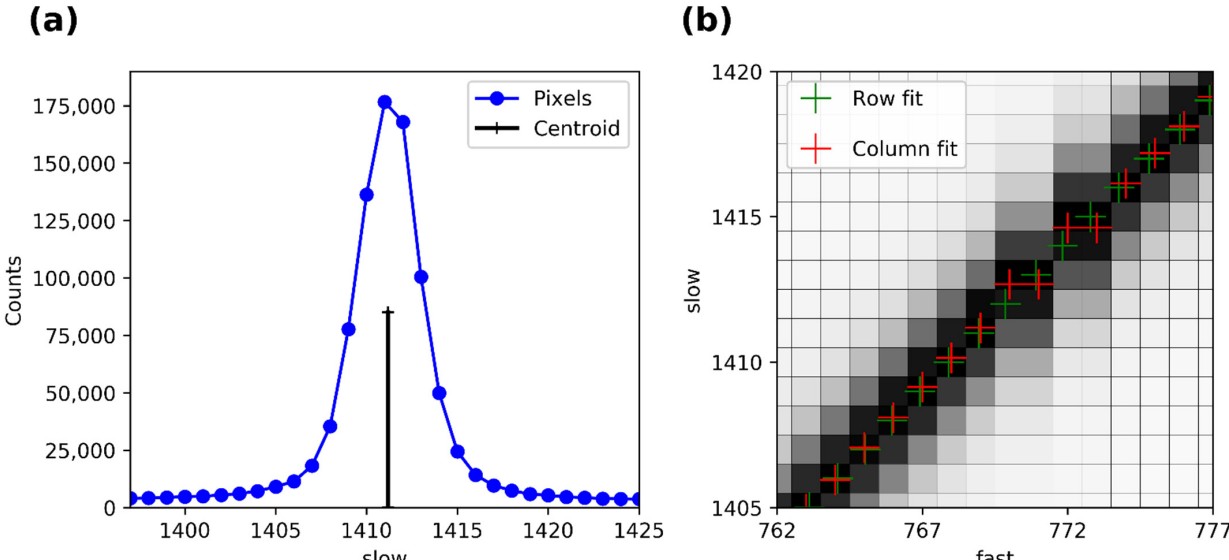

**Figure 2.** (**a**) Photon counts and peak position for a single column of pixels. (**b**) A region of interest on the detector where columns 770–773 show the effects of a virtual pixel correction at the edge of a physical readout chip. Slow and fast labels refer to the pixel directions in the image.

### 3.2. Geometric Refinement via Least Squares

The initial detector geometry was estimated and fitted using the (simplex) algorithm in the ImagD11 software [25] without any corrections for module alignment errors. This method was sufficient to assign peaks to *hkl* powder rings and to observe large steps at the module boundaries. Several problematic rows and columns of pixels were identified during the initial fitting. These were usually due to chip or module boundaries or other defects, and they are clearly seen as outliers in Figure 3b,c. All such problematic points were excluded from the fitting because the main priority here was to extract the overall module geometry. The large residuals observed at the end of the initial calibration (Figure 3b) without spatial correction gave a strong motivation to obtain a more precise description of the module geometry.

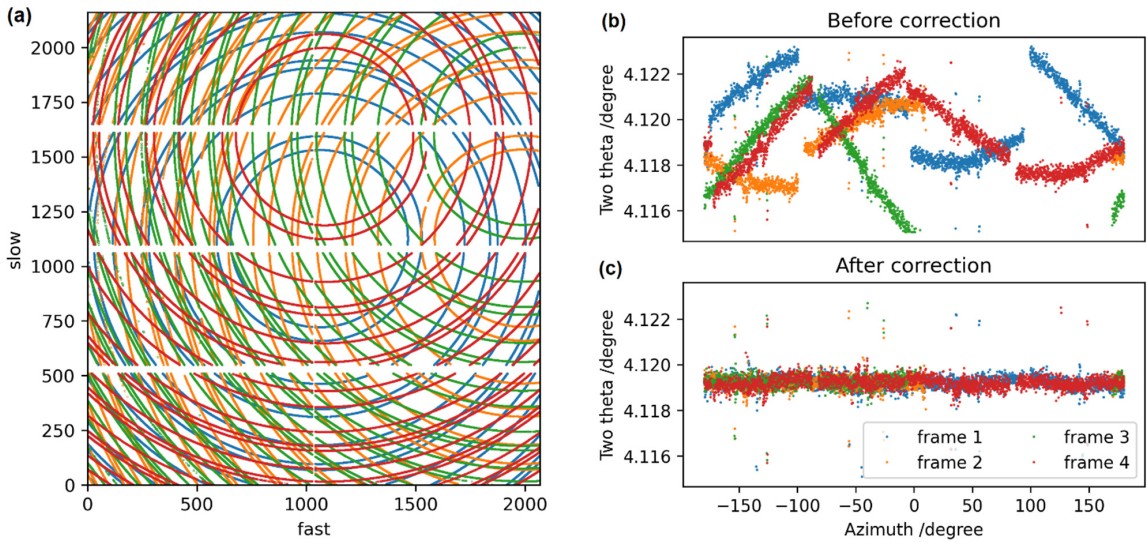

**Figure 3.** (**a**) Superposition of the Debye–Scherrer ring positions from the four calibration images where slow and fast refer to the pixel directions in the image. The variation of the Bragg angle (two theta) for the low angle (111) reflection from the $CeO_2$ powder data before (**b**) and after (**c**) applying the calibration.

The moving blocks of pixels were defined in terms of a 4 × 4 grid for this detector, and it was assumed that the four chips within a half-module were fixed onto the sensor with negligible errors and that all pixels are coplanar. Exact details of the module layout for other detectors will depend on the detector design, manufacturing, and assembly process. The geometry model was defined as a rotation around the center of the module, followed by a translation in the plane, apparently giving 16 × 3 = 48 parameters to fit. In order to remove singularities from the mathematical problem, the mean of the shift and rotation angle of all modules was set to zero by fixing the position and angle of one module as the sum of the previous 15, leading to 45 fitting parameters for the distortion model. We assumed that any out of plane component was negligible (this was not the case for the original Pilatus1M in ref. [26]), and in this setup, the data were insensitive to such shifts due to the large sample to detector distance. There are five geometrical parameters to be fitted for the detector geometry of each powder pattern, and these include distance (x), beam center (y/z), and tilts (y/z). Additional parameters for single-crystal data to describe the rotation axis geometry are not required here. It is not possible to determine all the distortion parameters using only one powder pattern because the image can be rotated around the X-ray beam axis. A unique minimum only emerges when two or more patterns are combined to produce a grid-like pattern in a single fit. For the four frames (Figure 1) used here, there are 4 × 5 = 20 parameters for the diffraction geometry in addition to the 45 for the distortion.

A minimization function to compute a misfit in terms of the relative squared length of the reciprocal lattice vectors was defined and then optimized using the curve fit function in the scipy package [27]. Inspection of the correlation matrix from the fit highlighted problems to determine the wavelength, detector tilts, and distance independently of each other. For the final refinements, the wavelength was set to a fixed value that was found after optimization and outlier data (0.76%) were removed. Although the computational problem was relatively large, with 65 variables fitted to 223,184 peaks, the computations were completed in less than a minute. The highest correlations were between distances and tilts, especially for the frames with the beam center close to the edge of the image. The least squares fitting errors for the module shifts were in the range $5 \times 10^{-4}$ to $2.7 \times 10^{-3}$ pixels (e.g., 37–200 nm) and with angle uncertainties of 1.9–2.8 μrad.

The mean absolute shifts in module positions were 0.164, 0.124 pixels with an angle of 0.42 mrad, while the maximum deviations were 0.34, 0.31 pixels and an angle of 1.1 mrad. The exact numerical values depend on the assembly of the individual detector, and we observed changes when a module was replaced by the manufacturer. For practical use of these results, the pixel shifts in the slow and fast array directions are stored as 2D images because these could be accepted by some of the software packages in use at the beamline, especially pyFAI [28] and ImageD11 [24].

### 3.3. Validation Using a Fine-Sliced Single-Crystal Dataset for Silicon

Fine-sliced single-crystal data were collected during a full rotation of 360° for a small silicon flake with 0.002° step size and 2-ms exposure time, recording 180,000 frames in 6 min. The uncompressed raw data might require about 1.5 TB of storage, but the frames data compress very well in bitshuffle-lz4 format [29], as most pixels are zero. The final dataset was only 48 GB, and the data rate was 133 MB/s, which can be written to disk during the scan. The sample to detector distance was about 20 cm, and the nominal X-ray energy was 44 keV with a band pass of dE/E ~0.002. This kind of data collection is intended to give a precise calibration of the X-ray energy (and band pass) based on the known unit cell parameter of silicon. The use of very fine slicing removes a class of uncertainties in the fitting, and it is, in principle, possible to fit the wavelength independently of the diffraction detector geometry (e.g., via a modification of the Bond method [30]). For our purposes here, we focus on validation of the detector geometry correction.

Peaks were extracted from the frames as 3D connected objects using the peak-searching code in the ImageD11 software package that computes their center of gravity positions. Af-

ter filtering peak to remove weak data and spots at the edge of the detector, 707 diffraction spots remained for parameter fitting. A single orientation was indexed using the known unit cell for silicon (space group Fd3m, 5.34094 Å). Refinement of the experimental geometry, crystal position, and orientation matrix were carried using the 3D spot positions both before and after the spatial distortion correction derived in Section 3.2 had been applied. No symmetry constraints were applied in order to use the deviation from cubic symmetry as an estimate of the systematic errors remaining in the experiment and calibration.

Figure 4 shows the residuals from the fit, which are expressed in terms of an error in reciprocal lattice vector. These values are the difference from integer values for the *hkl* indices that are computed from diffraction spot positions after conversion to scattering vectors and transformation to *hkl* using a fitted crystal lattice. In order to quantify the effect of this correction on the precision of the fitted lattice parameters, we computed a deviatoric strain tensor by comparison to a cubic crystal with the same volume and orientation. Before the distortion correction, the eigenstrain values were ($-13 \times 10^{-6}$, $-33 \times 10^{-6}$, and $46 \times 10^{-6}$), and after correction, these were reduced to ($9 \times 10^{-6}$, $12 \times 10^{-6}$, and $-21 \times 10^{-6}$). These values for strains are an order of magnitude lower than the numbers typically obtained for monochromatic diffraction experiments [31,32] and closer to the numbers achieved using Laue-DIC methods [33,34]. This improvement may be due to the use of extremely fine angular slicing combined with excellent counting statistics but perhaps also due to the use of a deviatoric strain tensor that neglects the wavelength uncertainties.

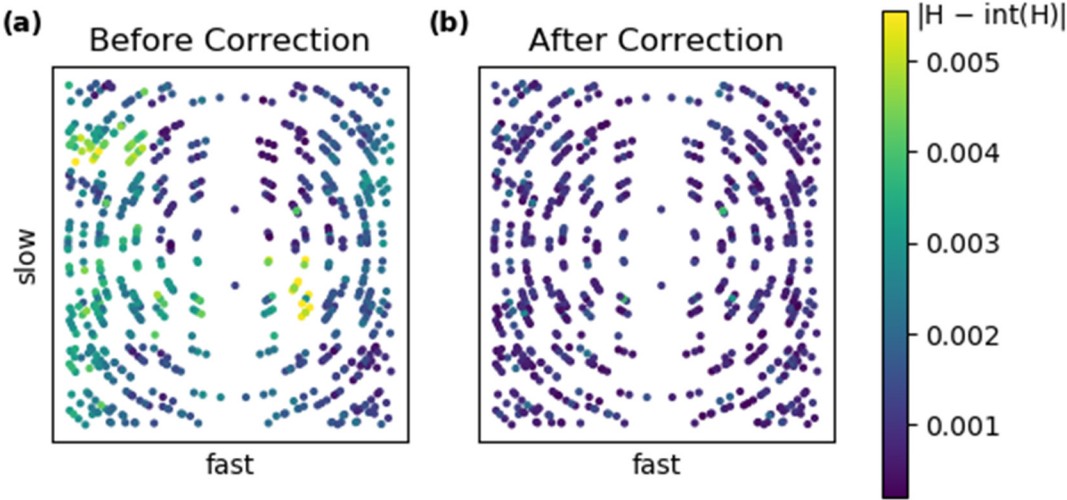

**Figure 4.** Plots of the deviations of spot positions from integer (*h,k,l*) for the silicon single crystal as a function of position on the detector face. (**a**) Data prior to the spatial correction derived from the powder data. (**b**) The same data after the correction derived from the powder data was applied.

## 4. Discussion

In comparison to the use of a standard grid object, the exploitation of diffraction data itself has the advantage of bringing the calibration phantom into the detector plane. The approach we used here was inspired by the methods based on single-crystal spot positions, with the key difference that we use a continuous feature in the powder images that can allow a calibration over all spatial frequencies. This offers a route to obtain a precise characterization of effects, such as charge sharing in module boundaries. The quality of the calibration obviously depends on the images used. We selected a small particle size for a strongly scattering material and ensured good counting statistics for this work. The procedure was originally developed with the expectation that it would be a one-off calibration, but this had to be repeated after a detector module was replaced by the manufacturer.

Currently, one of the limitations for using these detectors is the problem to export the distortion correction and detailed pixel-by-pixel geometry into a format suitable for

different integration software packages. When using the Esperanto format [35] for the crysalis software, for example, the software assumes that an image is already corrected for spatial distortion. With features like doubled pixels, it is not obvious how to produce a corrected image that will track statistics and pixel-by-pixel spatial distortion properly. The NeXus format and gold standard [36] do offer a route for crystallographers to include all these different details, and their more widespread adoption for chemical and physical crystallography may bring some benefits to the community in the future. If a detector is already installed on suitable translation stages, then it is fairly quick and easy measure a precise calibration via this powder method. Future implementations of this distortion correction approach into more user-friendly software packages should be a very welcome progress for the community.

**Author Contributions:** Conceptualization, investigations, and review and editing of the manuscript were carried out by J.P.W., C.G., and E.L.B.; software and the initial draft were written by J.P.W. All authors have read and agreed to the published version of the manuscript.

**Funding:** This research was funded by ESRF proposal BLC-12841.

**Data Availability Statement:** Jupyter notebooks are available in a github repository located at: https://github.com/jonwright/EigerPowderSpatial (accessed on 14 January 2022).

**Acknowledgments:** We thank the ESRF for beamtime under proposals BLC-13358 and BLC-12841. We are grateful to our colleagues Thomas Buslaps, Henri Gleyzolle, Eric Gagliardini, and Emmanuel Papillon for the technical, mechanical, and software installation of the detector. We thank Jerome Kieffer for providing an initial calibration based on an image of a grid that was itself kindly provided by Gavin Vaughan and beamline ID15a. We thank Marie Ruat for managing all aspects of the detector installation project as well as carrying out various acceptance tests and recording the grid image. We are grateful to the referees for their constructive feedback.

**Conflicts of Interest:** The authors declare no conflict of interest.

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
