# Peer review of "Using Powder Diffraction Patterns to Calibrate the Module Geometry of a Pixel Detector"

_crystals, doi:10.3390/cryst12020255_

Round 1
Reviewer 1 Report
Using powder diffraction data to refine module detectors is very useful to the community that desires high accuracy of pixel positions in a detector. The work is well presented and the authors make a good contribution by sharing their codes with the community.
There are a few suggestions that may help improve the quality:
- The introduction for current detector geometry optimization can be more comprehensive by including several other relevant work from either serial crystallography or powder diffractions.
- The performance of this method can be better assessed by comparing to other methods (if there are peers).
- It is an optimization problem in 45-dimension space, and whether the optimization guarantees a global minimum in such a high-dimension space? How robust is the algorithm when refine the parameters with partial data? (say, masking out 10% to 90% of peaks)
- What about the whole detector position/orientation? I think the total number of independent parameter is still 48, considering each module is independent.
Author Response
We thank the referee for their kind and constructive feedback. Responses to the specific points are below.
The performance of this method can be better assessed by comparing to other methods (if there are peers).
The discussion (section 4) briefly touches on this point but we prefer to focus on the validation described in section 3.3 because that is more representative for this instrument.
It is an optimization problem in 45-dimension space, and whether the optimization guarantees a global minimum in such a high-dimension space?
This is an interesting question. The dimension of the space does not create a problem provided the model fits the data and the distortion is small (the maths is approximately linear). We added the following sentences to section 3.2 to clarify this point:
“It is not possible to determine all the distortion parameters using only one powder pattern because the image can be rotated around the X-ray beam axis. A unique minimum only emerges when two or more patterns are combined to produce a grid-like pattern in a single fit. For the four frames (Figure 1) used here there are 4x5=20 parameters for the diffraction geometry in addition to the 45 for the detector distortion.”
How robust is the algorithm when refine the parameters with partial data? (say, masking out 10% to 90% of peaks)
As a check, the fits were repeated using a random selection of 10% of the peaks (rejecting 90% so that 22074 remained). The mean absolute shifts in module translations were 0.0023/0.0033 pixels and 6.2e-6 radians. These numbers validate the quoted least squares fitting errors after taking account of the reduction in the number of observations (we expected a sqrt(N) effect). Because this does not alter the results, we left this computation out of the manuscript.
We added a phrase to clarify that outliers are removed in the final fit (99.3% of data were used) because this makes the fit more robust (at the cost of removing a unique global minimum).
What about the whole detector position/orientation? I think the total number of independent parameter is still 48, considering each module is independent.
The geometry parameters were fitted. We hope that section 3.2 is now clearer.
Reviewer 2 Report
Dear authors,
it was a pleasure to review the manuscript. It is well prepared and structured. Please find the detailed report attached.
I am looking forward to your clarifications.

Author Response
We thank the referee for their kind and constructive feedback and for carefully locating lots of ways to improve the text. Responses to the specific points are given below.
All of the typographical points should have been corrected (except line 27 “the ESRF”).
The following sentences were added to the Materials and Methods section to describe the detector modules and gaps in more detail:
“A single image of 2068x2162 pixels is delivered to the user that includes mask values for gaps or other defects. The pixel size is 75µm square and the CdTe sensor thickness is 0.75 mm. The detector is assembled from 2x4 modules that each cover 77.1x38.4 mm with a vertical gap of 2 pixels in the center. Between these 2x4 modules, there is a 12 pixel vertical gap in the center of the detector and three horizontal gaps of 38 pixels. These gaps are shown in white on Figure 1.”
In section 3.1 the image segmentation method was clarified; the threshold of 5-sigma is indeed given by Poisson counting statistics for each pixel.
We have removed the informal comments about computing time (lines 121-122 and 172) to leave only the run times.
In section 3.2: figure 3 is re-labelled to have 3 panels (a,b,c). The detector description in the least squares paragraph was rephrased:
“The moving blocks of pixels were defined in terms of a 4 × 4 grid for this detector and it was assumed that the four chips within a half-module were fixed onto the sensor with negligible errors and that all pixels are coplanar.”
In section 3.3: Without compression we are unable to able to record such a fine sliced scan in a reasonable amount of time, we added a sentence to try to clarify this point:
“The final dataset was only 48 GB and the data rate was 133 MB/s, which can be written to disk during scans.”
Quantifying the wavelength uncertainty would lead to a circular argument here because we would normally use a silicon dataset like this for the calibration. The deviatoric strains suggest a relative error of the order ~3e-5.
For the references:
A link to the Dectris website is now listed as reference 1 for the detector description.
Reference 22 (now 23) was recommended by the jupyter authors at this weblink: https://github.com/jupyter/jupyter/issues/190
We have corrected reference 29 (now 30 for Bond, 1960).